# Surface Roughness Effects on the Vibration Characteristics of AT-Cut Quartz Crystal Plate

**DOI:** 10.3390/s23115168

**Published:** 2023-05-29

**Authors:** Mengjie Li, Peng Li, Nian Li, Dianzi Liu, Iren E. Kuznetsova, Zhenghua Qian

**Affiliations:** 1State Key Laboratory of Mechanics and Control for Aerospace Structures, College of Aerospace Engineering, Nanjing University of Aeronautics and Astronautics, Nanjing 210016, China; limengjie1993lh@163.com (M.L.); lipeng_mech@nuaa.edu.cn (P.L.); nianli@nuaa.edu.cn (N.L.); 2Shenzhen Research Institute, Nanjing University of Aeronautics and Astronautics, Shenzhen 518063, China; 3School of Engineering, University of East Anglia, Norwich NR4 7TJ, UK; dianzi.liu@uea.ac.uk; 4Kotel’nikov Institute of Radio Engineering and Electronics of RAS, Moscow 125009, Russia; kuziren@yandex.ru

**Keywords:** quartz crystal plate, surface roughness, mode coupling, activity dip

## Abstract

With the miniaturization and high-frequency requirements of quartz crystal sensors, microscopic issues affecting operating performance, e.g., the surface roughness, are receiving more and more attention. In this study, the activity dip caused by surface roughness is revealed, with the physical mechanism clearly demonstrated. Firstly, the surface roughness is considered as a Gaussian distribution, and the mode coupling properties of an AT-cut quartz crystal plate are systematically investigated under different temperature environments with the aid of two-dimensional thermal field equations. The resonant frequency, frequency–temperature curves, and mode shapes of the quartz crystal plate are obtained through the partial differential equation (PDE) module of COMSOL Multiphysics software for free vibration analysis. For forced vibration analysis, the admittance response and phase response curves of quartz crystal plate are calculated via the piezoelectric module. The results from both free and forced vibration analyses demonstrate that surface roughness reduces the resonant frequency of quartz crystal plate. Additionally, mode coupling is more likely to occur in a crystal plate with a surface roughness, leading to activity dip when temperature varies, which decreases the stability of quartz crystal sensors and should be avoided in device fabrication.

## 1. Introduction

The quartz crystal sensor is a highly sensitive micro and nano sensor with an AT-cut quartz crystal resonator (QCR) at its core. The quartz crystal resonator consists of quartz crystal plates that are equipped with electrodes on their top or bottom surfaces. Generally, the frequency characteristics of AT-cut crystal plates remain stable despite temperature variations, making them suitable as sensing devices [1,2,3]. Due to the smaller thickness of the AT-cut quartz plate, most resonators operate with thickness-shear (TSh) modes. However, due to wave reflection and superposition at the plate edges, pure TSh modes cannot exist in real devices and may be coupled with other modes. For instance, some spurious modes with their resonance frequencies extremely close to the resonant frequency are coupled with the TSh mode [4], such as extension (E) mode, face-shear (FS) mode, flexural (F) mode, etc. To avoid these unwanted modes and spurious modes, it is essential to obtain the aspect ratio of the quartz crystal plate [4,5,6,7]. As investigated in ref. [6], for the straight-crested waves and vibration modes around the operating TSh mode of the quartz crystal plate, it is necessary to exclude certain plate length/thickness ratios. However, most studies on this topic focus on the vibration characteristics of ideal resonators at room temperature [8,9,10,11,12,13,14]. It is still crucial to perform stability analyses of quartz crystal sensors under different temperatures to improve their operating performance.

To analyze the frequency stability of quartz crystal plates suffering from external temperature, it is necessary to establish the multi-field coupling theory that considers mechanical, electric, and temperature fields. To this end, Yong et al. derived three-dimensional thermal incremental field equations based on temperature coefficients of quartz obtained by Koga, Bechmann, Kahan, and others [15,16,17,18,19,20]. By using these equations, the frequency shift caused by temperature variation can be predicted. Zelenka analyzed the effects of electrodes on the frequency–temperature characteristics of quartz resonators, when the quartz piezoelectricity is approximately ignored [21]. Sekimoto et al. theoretically and experimentally analyzed the influence of temperature on other spurious frequencies near the fundamental TSh mode in quartz crystal plate [22]. Recently, modified scalar equations for circular resonators that consider thermal effects were derived in ref. [23]. However, these studies only conducted thermal stability analysis on a quartz crystal plate, without considering the activity dip when external temperature varies.

In QCR, rapid frequency shifts occur in certain temperature ranges, which are known as “activity dips” [24]. Koyama’s experimental results revealed that rapid frequency shifts are responsible for couplings between the fundamental TSh mode and nearby spurious modes, which further lead to activity dips in AT-cut resonators [25]. This experiment is only conducted at room temperature. In fact, the frequency positions of mode coupling are temperature-dependent in contoured quartz resonators, which can easily cause frequency shift [26,27]. Compared with experimental attempts, finite element analysis is considered as the most widely used method to analyze activity dips and predict spurious modes in resonators and sensors [28,29]. Because of the detrimental effects on resonators, the activity dip should be avoided, and effective methods have been proposed, such as adjusting the shape or crystal size. With the aid of finite element analysis, it has been demonstrated that the activity dip is related to the structural shape and size and should be avoided.

In practice, the sensor’s surface is not ideally smooth. With the development of device miniaturization, mismachining tolerances that were previously neglected are now being revealed, and the surface roughness of the crystal becomes one of the key issues [30,31]. All surfaces contain some roughness, which can destroy the original structure and the symmetry of microscopic devices if the surface roughness is relatively high. Experimental results show that surface roughness has a significant impact on the resonant frequency and Q-factor of QCR [32,33]. This was also shown in the case of a contact quartz crystal sensor with liquid [34,35]. Therefore, this study aims to investigate the effect of surface roughness on the vibration characteristics of a quartz crystal plate and proposes reasonable avoidance or compensation measures to improve the device stability, which is the highlight of the present contribution.

This study presents a systematic investigation of the coupled vibration analysis of a quartz crystal plate based on the thermal incremental field theory, in which the crystal surface roughness is included. To reveal the activity dip caused by surface roughness, we utilized the PDE and piezoelectric modules of the COMSOL Multiphysics software to analyze the frequency–temperature characteristics and admittance response, respectively. The conclusions drawn from this study can provide a fundamental understanding of the coupling vibrations under thermal temperature, and are of guiding significance for designing quartz crystal sensors with enhanced stability.

## 2. Theoretical Analysis of a Quartz Crystal Plate

For the convenience of theoretical analysis, the surface roughness is usually treated as periodically distributed [36], e.g., cosine, triangular or rectangular function. However, many structure surfaces are randomly rough, with a large number of asperities, and the asperity height belongs to a Gauss distribution [37]. Zhang et al. [38] verified that the surface height with a Gauss distribution is closer to the real appearance of the crystal surface. Therefore, the Gauss distribution is utilized to describe the random surface in this paper. A quartz plate with surface roughness is considered, and the surface roughness is characterized by the peak height of asperities, *H_m_*, as in Figure 1. The plate thickness is *h_q_*, and $h_{e}^{1}$ and $h_{e}^{2}$ are the thickness values of upper and bottom electrodes, respectively. The quartz plate length is *l_q_*, and the electrode length is represented by *l_e_*, respectively. The aim of this contribution is to investigate the influence of surface roughness on the coupling vibration under different temperature environments.

The influence of surface roughness will be revealed by the theoretical analysis and numerical simulations. Firstly, three-dimensional thermal incremental field equations derived by Lee et al. [39,40] are employed to analyze the vibration properties of quartz crystal plate, which is the basis of the theoretical analysis in this paper. Owing to a large length/thickness ratio of the quartz crystal plate, the two-dimensional section model considering the straight-crested wave propagating in the *x*_1_ direction is chosen. *x*_2_ is along the plate thickness, and *x*_3_ is determined from *x*_1_ and *x*_2_ by the right-hand rule, such as Figure 1. Thus, the displacement solution is independent of *x*_3_, i.e., ∂/∂*x*_3_ = 0. Therefore, the two-dimensional thermal incremental field equations are reduced as
(1)$$\begin{array}{l} \beta_{11}\left(T_{11,1}+T_{21,2}\right)=\rho {\ddot{u}}_{1},\\ \beta_{22}\left(T_{21,1}+T_{22,2}\right)+\beta_{23}\left(T_{31,1}+T_{32,2}\right)=\rho {\ddot{u}}_{2},\\ \beta_{32}\left(T_{21,1}+T_{22,2}\right)+\beta_{33}(T_{31,1}+T_{32,2})=\rho {\ddot{u}}_{3},\\ D_{1,1}+D_{2,2}=0, \end{array}$$
where *T_ij_* and *D_i_* (*i*, *j* = 1, 2, 3) represent the incremental stress and the incremental electric displacement, respectively. *ρ* is the mass density, and *u_j_* denotes the incremental displacement. The index after a comma denotes partial differentiation with respect to the coordinate, and the dot on the symbol above denotes the differential operation with respect to time.

According to the piezoelectric Lagrangean equations for the frequency–temperature behavior of quartz resonators derived by Yong and Wu, the incremental displacement in Equation (1) is controlled by the following geometric equations [41]
(2)$$\begin{array}{l} S_{11}=\beta_{11}u_{1,1},\;S_{22}=\beta_{22}u_{2,2}+\beta_{23}u_{3,2},\\ S_{32}=\beta_{23}u_{2,2}+\beta_{33}u_{3,2},\;S_{31}=\beta_{23}u_{2,1}+\beta_{33}u_{3,1},\;\\ S_{12}=\beta_{11}u_{1,2}+\beta_{23}u_{3,1}+\beta_{22}u_{2,1},\\ E_{1}=-\phi_{,1},\;E_{2}=-\phi_{,2}. \end{array}$$

The corresponding constitutive equations are [41]:(3)$$\begin{array}{l} T_{11}=C_{11}^{\theta }S_{11}+C_{12}^{\theta }S_{22}+C_{14}^{\theta }S_{32}-e_{11}^{\theta }E_{1},\\ T_{22}=C_{21}^{\theta }S_{11}+C_{22}^{\theta }S_{22}+C_{24}^{\theta }S_{32}-e_{12}^{\theta }E_{1},\\ T_{32}=C_{41}^{\theta }S_{11}+C_{42}^{\theta }S_{22}+C_{44}^{\theta }S_{32}-e_{14}^{\theta }E_{1},\\ T_{31}=C_{55}^{\theta }S_{31}+C_{56}^{\theta }S_{12}-e_{25}^{\theta }E_{2},\\ T_{12}=C_{65}^{\theta }S_{31}+C_{66}^{\theta }S_{12}-e_{26}^{\theta }E_{2},\\ D_{1}=e_{11}^{\theta }S_{11}+e_{12}^{\theta }S_{22}+e_{14}^{\theta }S_{32}+\varepsilon_{11}^{\theta }E_{1},\\ D_{2}=e_{25}^{\theta }S_{31}+e_{26}^{\theta }S_{12}+\varepsilon_{22}^{\theta }E_{2}. \end{array}$$
where *S_ij_* and *E_i_* represent the strain and electric field, respectively, and ϕ is the incremental potential function. In Equations (1)–(3), the elastic constant $C_{pq}^{\theta }$, piezoelectric coefficient $e_{pq}^{\theta }$, dielectric constant $\varepsilon_{pq}^{\theta }$, and thermal expansion coefficient *β_ik_* (*p*, *q*, *k* = 1,2,3) are expressed as [19]
(4)$$\begin{array}{l} C_{pq}^{\theta }=C_{pq}+C_{pq}^{\left(1\right)}\theta +C_{pq}^{\left(2\right)}\theta^{2}+C_{pq}^{\left(3\right)}\theta^{3},\\ e_{pq}^{\theta }=e_{pq}+e_{pq}^{\left(1\right)}\theta +e_{pq}^{\left(2\right)}\theta^{2}+e_{pq}^{\left(3\right)}\theta^{3},\\ \varepsilon_{pq}^{\theta }=\varepsilon_{pq}+\varepsilon_{pq}^{\left(1\right)}\theta +\varepsilon_{pq}^{\left(2\right)}\theta^{2}+\varepsilon_{pq}^{\left(3\right)}\theta^{3},\\ \beta_{ik}=\delta_{ik}+\alpha_{ik}^{\left(1\right)}\theta +\alpha_{ik}^{\left(2\right)}\theta^{2}+\alpha_{ik}^{\left(3\right)}\theta^{3}. \end{array}$$
with the temperature change *θ* = Δ*T* = (*T* − *T*_0_). The reference temperature *T*_0_ is set to 25 °C. *δ_ik_* represents the Kronecker delta, and $\alpha_{ik}^{\left(n\right)}$ denotes the *n*th-order temperature coefficients of thermal expansion.

The boundary conditions on the top surfaces are:(5)$$\begin{array}{l} T_{12}=-\rho^{\prime }h_{e}^{1}u_{1,tt},T_{22}=-\rho^{\prime }h_{e}^{1}u_{2,tt},\;\phi =+(0.5V)\exp(i\omega t),\;\;\;\;\;\;\;\;\;\left|x_{1}\right|\leq 0.5l_{e}\\ T_{12}=0,\;T_{22}=0,\;\;\;\;\;\;\;D_{2}=0,\;\;\;\;\;\;\;\;\;\;\;\;\;\;\;\;\;\;\;\;\;\;\;\;\;\;\;\;\;\;\;\;\;\;\;\;\;\;\;\;\;\;0.5l_{e}\leq \left|x_{1}\right|\leq 0.5l_{q} \end{array}$$
and on the bottom surfaces they are:(6)$$\begin{array}{l} T_{12}=+\rho^{\prime }h_{e}^{1}u_{1,tt},T_{22}=+\rho^{\prime }h_{e}^{1}u_{2,tt},\;\phi =-(0.5V)\exp(i\omega t),\;\;\;\;\;\;\;\;\;\left|x_{1}\right|\leq 0.5l_{e}\\ T_{12}=0,\;T_{22}=0,\;\;\;\;\;\;\;D_{2}=0,\;\;\;\;\;\;\;\;\;\;\;\;\;\;\;\;\;\;\;\;\;\;\;\;\;\;\;\;\;\;\;\;\;\;\;\;\;\;\;\;\;\;0.5l_{e}\leq \left|x_{1}\right|\leq 0.5l_{q} \end{array}$$

*V* is the driving voltage, and ω is the driving frequency. For free vibration, *V* is set to zero to achieve a homogeneous solution, while for the forced vibration, we set *V* = 3 V.

At the left and right edges of the plate, the traction-free and electrical open circuit boundary conditions require [5]:(7)$$T_{21}=0,\;T_{11}=0,\;\;\;\;\;\;\;D_{1}=0.\;\;$$

For the forced vibration of the quartz plate, the electric field *E*_2_ is related to the driving voltage *V* [6]:(8)$$E_{2}=-V\exp(i\omega t)/h_{q}.$$

According to the constitutive Equation (3), the relationship between the electric dis-placement in the *x*_2_ direction and the strain and driving voltage can be obtained. Therefore, the relationship between the total charge *Q* on the top electrode and the potential displacement can be obtained [12]:(9)$$Q=-2w_{e}\int_{-l_{e}}^{l_{e}}D_{2}dx_{1}.$$

The equivalent current *I* is the time derivative of charges induced on the surfaces of the plate [42]:(10)$$\;I=\dot{Q}=i\omega Q.$$

After that, the admittance of the quartz plate per unit electrode area can be calculated via [12]:(11)$$Y=I/V/(4l_{e}w_{e}),$$
where *w_e_* is the width of the electrode.

In summary, the coupled vibration in a crystal plate with surface roughness is controlled by the governing Equation (1). For free vibration, the boundary conditions are Equations (5)–(7), whilst Equations (8)–(11) need to be added to solve the working performance when the plate is excited by external voltage imposed on the electrodes. In the following section, this paper will solve the free vibration and forced vibration, respectively, and systematically investigate the influence of surface roughness.

## 3. Numerical Results and Discussions

In general, obtaining a theoretical solution that simultaneously satisfies both the governing Equation (1) and the boundary conditions (5)–(7) is challenging, especially for a plate with the consideration of surface roughness. In order to obtain the vibration properties of a quartz crystal plate, the PDE module of the COMSOL Multiphysics software is employed to solve the problem. To characterize a random geometric surface, the parametric surface geometric features provided by COMSOL Multiphysics software are used to generate a synthesized random surface. After adding a parametric surface node, the height distribution can be determined by the peak height *H_m_*.

Subsequently, the material parameters, such as the *n*th-order temperature derivatives of the elastic constants, piezoelectric constants, dielectric constants, and coefficient of thermal expansion, are input into the PDE module of COMSOL Multiphysics software along with Equations (2) and (3). The boundary conditions in COMSOL Multiphysics software are then set according to Equations (5)–(7), where the inertial effect of the electrodes is considered and replaced by the additional mass layer. Finally, mesh is distributed based on the frequency range, and the eigenfrequency is determined after ensuring convergence. This study considers the AT-cut quartz crystal layer with *l_q_* = 1.2 mm, *h_q_* = 0.03 mm, and electrode layers with *l_e_* = 0.7 mm, *w_e_* = 0.4 mm, $h_{e}^{1}$ = 0.1 × 10^−3^ mm and $h_{e}^{2}$ = 0.2 × 10^−3^ mm. To improve calculation accuracy, a mapped distribution is employed in this study, and the plate thickness direction is divided into 20 elements, resulting in a total of 12,020 elements.

### 3.1. Free Vibrations Analysis

Firstly, the influence of surface roughness of AT-cut quartz plate on the frequency characteristics is considered. Figure 2 illustrates the relationship between frequency and the length/thickness ratio *l_q_/h_q_* at room temperature under different surface roughness conditions. The curves represent various data points which indicate the frequency of a given mode. Typically, the curves are divided into two families, the relatively flat and the relatively oblique. The resonant frequency of 53.213 MHz, represented by the flat portions, signifies the first-order thickness-shear mode (TSh1 mode) of the AT-cut quartz crystal plate, which is also the working mode. The spurious modes are represented by the other flat portions, e.g., the second TSh mode of 53.614 MHz. Additionally, there are other modes, such as the E, FS and F modes, represented by oblique portions. When the length/thickness ratio is close to the flat portions, e.g., *l_q_/h_q_* is located in the region of 37.2~37.8 or 38.1~38.7, the mode coupling effect is weak and the vibration of TSh1 dominates in quartz crystal plate. However, when the length/thickness ratio is close to the region where the flat and oblique portions intersect, e.g., *l_q_/h_q_* is located in the region of 37.98~38.02 or 40.46~40.62, the mode coupling effect is strong, and other modes such as the E, FS or F modes affect the TSh1 mode obviously. Consequently, it is vital to prevent the structural size of quartz crystal sensors from falling into the region where the flat and oblique portions intersect, meaning that the mode coupling effect is avoided as far as possible. Intuitively, Figure 2 demonstrates that surface roughness has a significant impact on the resonant frequency of the quartz crystal plate, and the resonant frequency decreases as the peak height of the surface roughness increases. This is due to fact that the increase in *H_m_* increases the thickness of the crystal plate, and thus reduces its frequency. Therefore, it is anticipated that the influence of surface roughness on the resonant frequency will become more and more evident when the plate thickness decreases.

To analyze the impact of surface roughness on mode coupling, a zoomed-in image of the frequency spectra in the green region in Figure 2 is presented in Figure 3. In Figure 3a, the TSh1 mode intersects the E and F modes, but there is no coupling when *H_m_* = 0. However, as shown in Figure 3b,c, when the crystal plate surface is rough, the flat portions are interrupted by oblique portions, resulting in strong coupling between the TSh1 mode and E or F mode. The comparison reveals that crystal plate surface roughness results in mode coupling, with greater *H_m_* causing stronger coupling. Because of surface roughness, the crystal plate is not symmetric along the *x*_2_ direction anymore, and then the E or F mode easily couple with the TSh1 mode.

To clearly demonstrate the effect of surface roughness on the mode coupling of quartz crystal sensors, the frequency–temperature is calculated and shown in Figure 4. Comparing Figure 4b,c with Figure 4a, the activity dip phenomenon of the frequency shifts can be seen if *H_m_* is not zero, and the temperature region of *H*_m_ = 600 nm related to activity dip is larger than that of *H_m_* = 300 nm. In order to further verify the above results, the mode shapes at Δ*T* = 10 °C and Δ*T* = 60 °C when *H_m_* = 300 nm are given and shown in Figure 5. In Figure 5a, there is an activity dip when Δ*T* = 10 °C, and the mode coupling effect is strong. Obviously, the energy has spread out of the electrode region. However, when Δ*T* = 60 °C, the mode coupling effect is weak, and energy trapping is evident. Therefore, no activity dip occurs in the quartz crystal plate. Reviewing Figure 3 and Figure 4, for an ideal quartz crystal sensor without surface roughness, i.e., *H_m_* = 0, the length/thickness ratio can be designed as 40.46~40.62, because the E and F modes decoupled from TSh1 and no activity dip occurs. However, if the surface roughness is generated, this ratio region is no longer suitable because of the activity dip.

Overall, surface roughness is harmful for quartz crystal sensors according to the results above, and should be avoided or minimized during manufacturing. To quantitatively validate the results obtained and qualitatively verify the mechanism revealed, forced vibration analysis will be conducted in the next sub-section by using the piezoelectricity module of COMSOL Multiphysics software.

### 3.2. Forced Vibration Analysis

In this section, the influence of surface roughness on the electrical response of a quartz crystal plate undergoing forced vibration is examined. A pair of driving voltages of ±3 V is applied on the top and bottom surfaces of the crystal plate, enabling the acquisition of the admittance response at various frequencies. Taking *l_q_*/*h_q_* = 40 as a typical case, the admittance response and corresponding phase distribution are calculated via Equation (11), and shown in Figure 6a,b, respectively.

Point A in Figure 6 can be determined as the essential TSh1 mode because the admittance reaches the maximum at this frequency, while points B and point C represent spurious modes. It can be found that the resonant frequency of the TSh1 mode decreases with the increase in *H_m_*. These frequencies are consistent with the results in Figure 2, which can validate the correctness of the numerical simulations in this paper. The maximum values of admittance and phase at the resonance frequency also decrease with increasing *H_m_*, as the surface roughness affects the thickness-shear vibration of the resonator, causing the energy concentrated in the electrode region to decay, as shown in Figure 7. Taking point B in Figure 6a as an example, when *H_m_* = 0, i.e., the surface of the crystal plate is smooth without surface roughness, there is no obvious electrical response of the spurious mode. However, when the surface of the crystal plate is not smooth, the symmetry of the quartz crystal plate along the thickness direction is destroyed by the surface roughness, and the displacement *u*_1_ is no longer symmetrical about the middle plane of the plate, as shown in Figure 8b,c. The positive and negative charges generated by the piezoelectric effect cannot completely be cancelled out, leading to ripples in the admittance and phase curves at the corresponding positions in Figure 6a,b, respectively.

To better illustrate the energy distributions of the sensor, Figure 9a,b depict the distributions of vibration energy in the middle region of the quartz crystal plate, extending to both boundaries. Specifically, Figure 9a,b display the distribution of the displacement component *u*_1_ along the *x*_1_ direction at x_2_ = 0.5 *h_q_* for two scenarios: no mode coupling of the quartz plate (*l_q_/h_q_* = 40) in Figure 9a and strong mode coupling of the quartz plate (*l_q_/h_q_* = 40.7) in Figure 9b, respectively. These figures indicate that crystal surface roughness impacts the energy position in the electrode region and accelerates the decay of vibration amplitude in the middle region of the quartz crystal plate. Moreover, the amplitude variation of *u*_1_ in Figure 9b is more complex, which is attributable to the crystal surface roughness. Therefore, to avoid frequency shifts, the length/thickness ratio of the resonator should be altered by grinding or the surface roughness should be improved by polishing.

## 4. Conclusions

Based on the two-dimensional thermal increment field equations, the surface roughness effect on the free and forced vibrations of quartz crystal plate are systematically investigated with the aid of COMSOL Multiphysics software. The surface roughness is expressed in the form of random distribution, which is closer to the real devices. After examining the temperature stability, it is found that the structural symmetry is broken by the surface roughness, which further increases the possibility of mode coupling and thus leads to activity dips. The process of energy capture within a quartz crystal plate has been observed through the analysis of its mode shape and displacement distribution, allowing for a concise depiction of the sensor’s vibration.

(1)The resonant frequencies, frequency–temperature curves and vibration modes of the quartz crystal plates are investigated via the free vibration analysis. It is shown that the crystal surface roughness reduces the operating frequency of resonators and further causes mode coupling, which is the primary reason for the activity dip in the resonator when subjected to temperature variations.(2)For forced vibration analysis, the admittance response and phase response curve of the quartz crystal plate are calculated through the piezoelectricity module. It is shown that surface roughness decreases the admittance and phase values of resonators. When the crystal surface is rough, the positive and negative charges generated by the piezoelectric effect cannot be completely balanced, resulting in the generation of ripples in the admittance and phase values, which is not conducive to the operation of resonators in the circuit.

Overall, the surface roughness is not beneficial for a device’s temperature stability, and should be avoided and minimized during device manufacturing and the service process. Not limited by quartz crystal sensors, the method and results presented in this paper can also be applied to other related devices, such as resonators, quartz crystal microbalances, and other similar devices, which are expected to be validated in experiments in the near future. This method can be widely used to study resonators at various frequencies. In addition, the temperature range in which the device experiences an activity dip can be determined, which can guide the appropriate operating temperature range for the device.

## Figures and Tables

**Figure 1 sensors-23-05168-f001:**
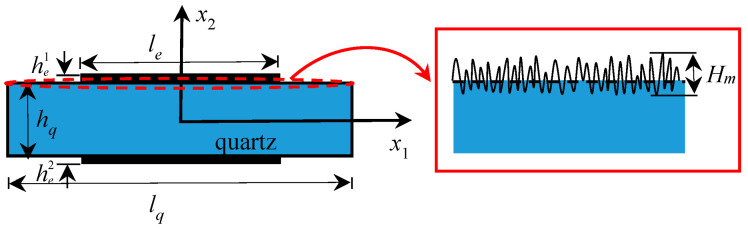
A quartz crystal plate with surface roughness.

**Figure 2 sensors-23-05168-f002:**
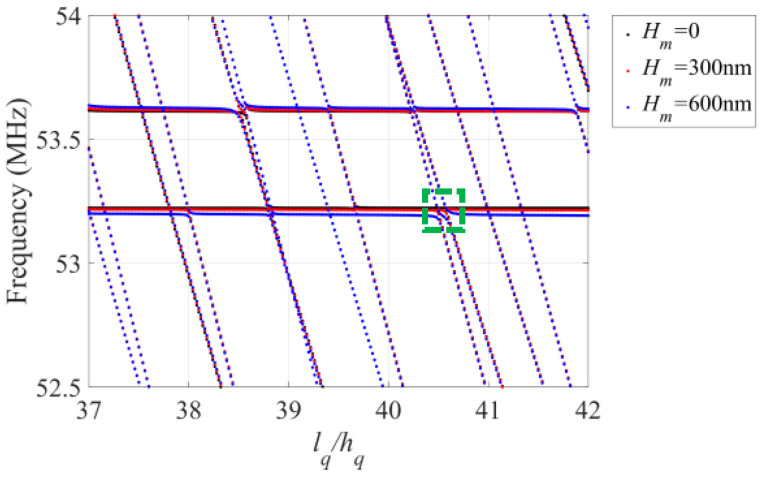
The frequency spectra at room temperature under different peak heights *H_m_*. The green dashed box in Figure 2 corresponds to the local frequency spectra in Figure 3.

**Figure 3 sensors-23-05168-f003:**
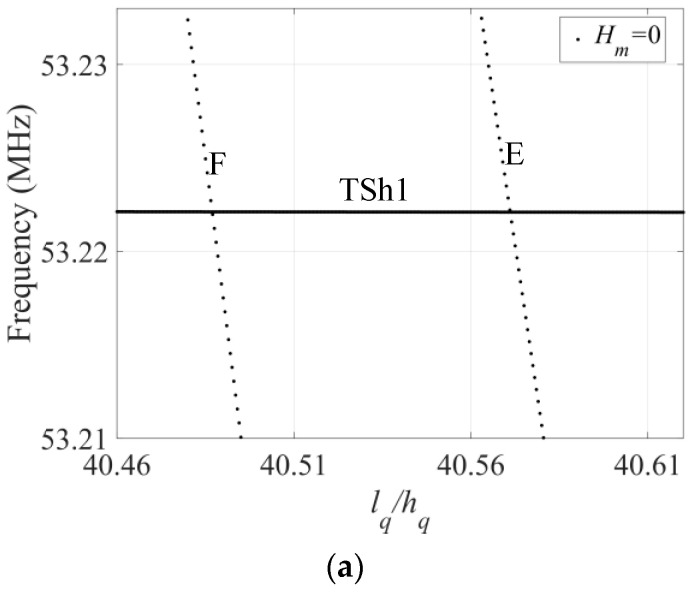
The local frequency spectra under different *H_m_*: (**a**) *H_m_* = 0; (**b**) *H_m_* = 300 nm; (**c**) *H_m_* = 600 nm.

**Figure 4 sensors-23-05168-f004:**
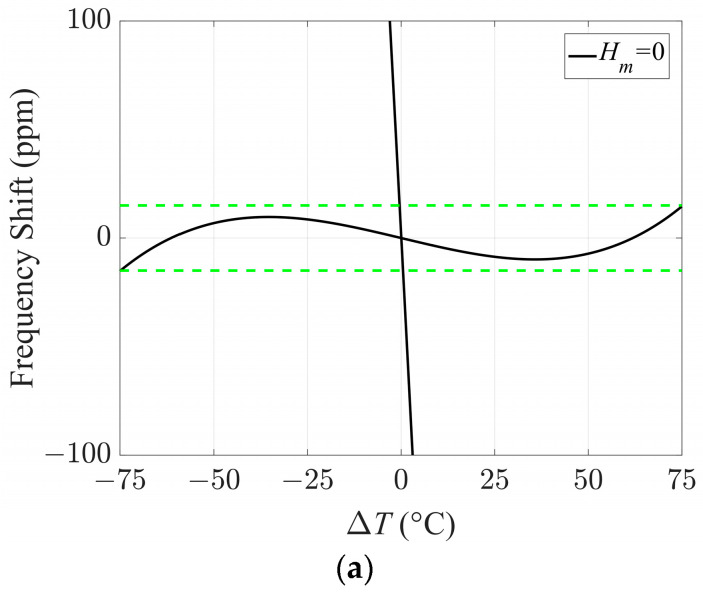
The frequency-temperature curves: (**a**) *H_m_* = 0 (the green dashes in Figure 4a represent the frequency tolerance range); (**b**) *H_m_* = 300 nm; (**c**) *H_m_* = 600 nm.

**Figure 5 sensors-23-05168-f005:**
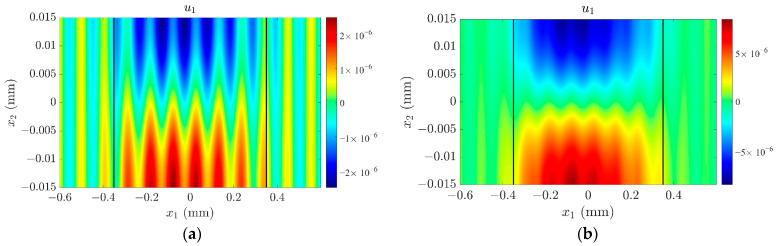
The mode shapes when *H_m_* = 300 nm: (**a**) Δ*T* = 10 °C; (**b**) Δ*T* = 60 °C.

**Figure 6 sensors-23-05168-f006:**
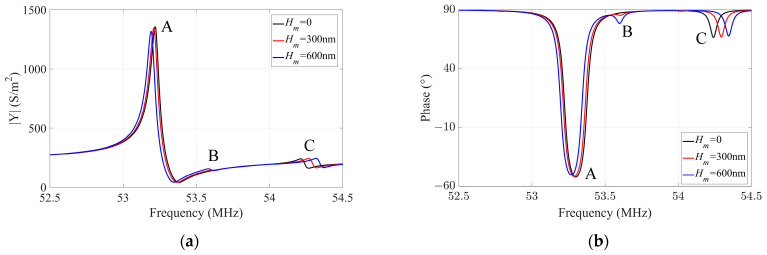
The response curves with different *H_m_*: (**a**) admittance response; (**b**) phase response.

**Figure 7 sensors-23-05168-f007:**
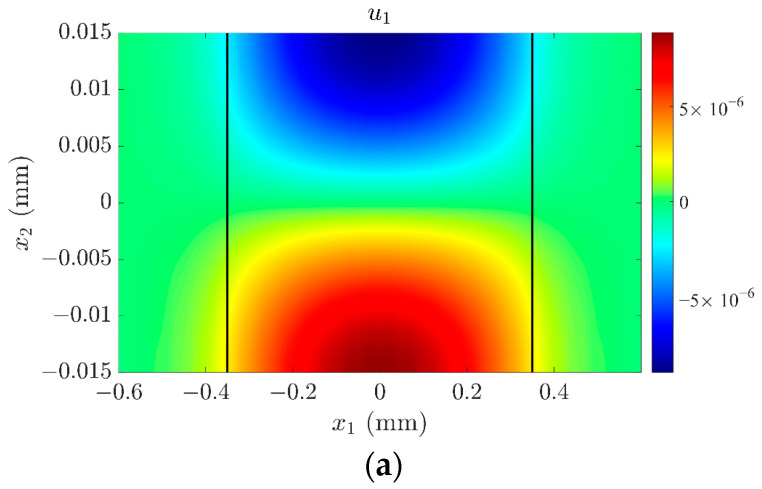
The mode shape of the TSh1 mode corresponding to admittance peak point A: (**a**) *H_m_* = 0; (**b**) *H_m_* = 300 nm, (**c**) *H_m_* = 600 nm.

**Figure 8 sensors-23-05168-f008:**
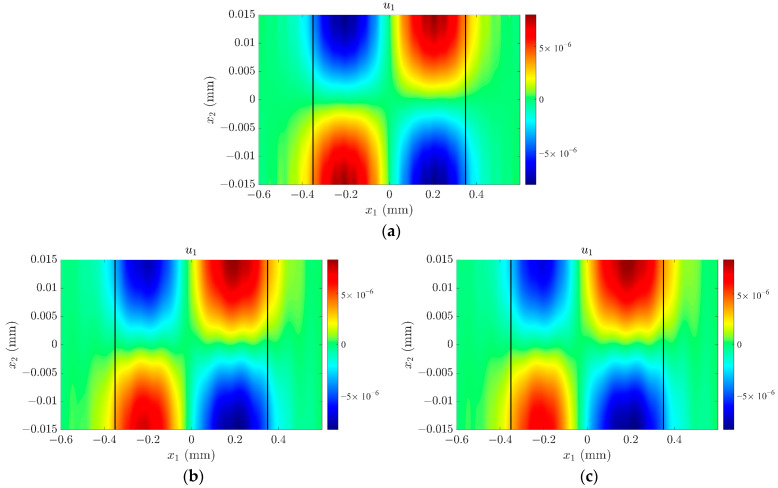
The mode shape of the spurious mode corresponding to admittance peak point B: (**a**) *H_m_* = 0; (**b**) *H_m_* = 300 nm, (**c**) *H_m_* = 600 nm.

**Figure 9 sensors-23-05168-f009:**
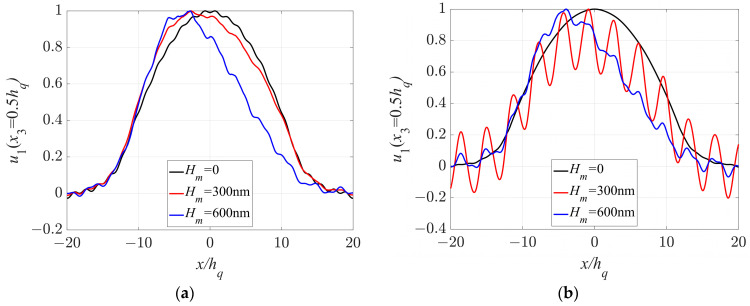
Distributions of *u*_1_ along the *x*_1_ direction at x_2_ = 0.5 *h_q_*: (**a**) *l_q_/h_q_* = 40; (**b**) *l_q_/h_q_* = 40.7.

## Data Availability

Not applicable.

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
