# Peer review of "Surface Roughness Effects on the Vibration Characteristics of AT-Cut Quartz Crystal Plate"

_sensors, 2023, doi:10.3390/s23115168_

Round 1
Reviewer 1 Report
This work investigated the surface roughness effects on vibration characteristics of AT-Cut Quartz Crystal Plate. It is an interesting work, but we found there are many researches about the influence of surface roughness, coupling mode, and temperature on the performances of quartz crystal, such as doi:10.1109 SPAWDA48812.2019.9019282, DOI: 10.1088/1674-1056/22/4/047702,doi:10.1063/1.338102, literatures [22], [23], [32],etc. Also, the author obtained results are common-sense conclusions, and some of which can be achieved from these above published works.Therefore, the author should give a detail illustrations about the advantages,distinction,and contributions of this work. Additionly,some formula deduction should provide a detail procedure. some english mistatakes should be modified, such as line 191,page 5.
The Quality of English Language is Ok especially for some minor mistakes.
Author Response
Response: The previous works mentioned above may be about surface roughness, coupling mode, or temperature, which are not related to activity dip of quartz crystal resonators when temperature changes. The present manuscript aims to reveal the activity dip caused by the surface roughness when temperature varies, which should be considered in the structural design before fabrication. The advantages, distinction, and contributions of this work have been demonstrated in detail, which are labeled on Page 11. This method can be widely used to study resonators at various frequencies. In addition, the temperature range in which the device experiences activity dip can be determined, which can guide the appropriate operating temperature range for the device. Additionally, more equations are added on Pages 3, 4 and 5, and clerical errors are also revised (Page 7).
Reviewer 2 Report
Dear Author(s), please find some comments on the manuscript ‘Surface Roughness Effects on Vibration Characteristics of AT-Cut Quartz Crystal Plate’, Manuscript ID: sensors-2329044:
1. From the ‘Abstract’ section there is not clear what proposes the manuscript, especially to which studies reflect to. The main idea of the Communication must be highlighted and mentioned.
2. The reflection fot the cited imtes newelty is not unified in the ‘Introduction’ section. In some cases the Author(s) present some studies, in other results and, respectively, for further some weak or strong parts in the current state of knowledge. Please try to refer to each references with the same wages, critical and pointing out some deficiencies.
3. The citation of various scholars is not unified as well. Some are referenced by surnames, some items by number, like ‘in Ref. [4]’, line 35. It is not written in a clear way.
4. Each of the cited paper should be introduced separately, respectively to the ‘Inbtroduction’ section again, e.g., [4-7], line 39, [8-14], line 43 or [15-20], line 49. In the last exemple nowelty and critical review should be placed for each of the researchers separately, indicating their progress in the studies.
5. The conclusions received from the ‘Introduction’ section are not derived from the critical review of the literature but, respectively, by the introducing some already finished studies. Proposal of further studies cannot be justified only by previous results but the lack in the current stage of knowledge. From that matter, the further studies must be fulfilling some current requirements, not only another similar studies.
6. All of the equations placed in the second section, titled ‘Theoretical Analysis of Quartz Crystal Plate’, are not introduced as newly proposed but are not referenced. It is not clear what is novelty and what was already presented. The Author(s) should separate, or even highlight, what they suggest and propose, respectively. From that point and comments no. 4 and 5, the motivation, even exist, is lost.
7. Considering the above (previous) comment, if Author(s) present any modification of the mathematical formulas, it must be indicated. Currently, the reader is lost and not receiving what the Author(s) are trying to convey.
8. In the third section, considering especially 3.1 and 3.2 subsetions, there is no discussion in fact. Any disadvantages and weak points of the studies presented is not introduced. What about the time of computation? Omly some general discussion problems, like in ‘The positive and negative charges generated by the piezoelectric effect cannot completely cancel out, leading to ripples in the admittance and phase curves at the corresponding positions in Figures 6a and 6b, respectively.’, lines 231-233, are placed. Please try to emphasize the advantages and disadvantages of the received results in a more clear way.
9. The ‘Conclusion’ section is not acceptable that, respectively, should be divided into separated and numbered gaps. The main (final) proposal must be highlighted and separated from thoe more in detail. Currently, this section is compose by various information not completely reflecting to the presention in the body part of the manuscript.
10. The full DOI links should be added to make the reader fast possibility to find the primary sources of the previous results.
Concluding, the manuscript reviewed seems to be interesting, and the area of study is up-to-date, definitely, nevertheless, some significant improvements must be provided before further processing of the manuscript reviewed, if allowed by the Editor that, respectively, in the current form, the reviewed manuscript cannot be accepted.
Author Response
- From the ‘Abstract’ section there is not clear what proposes the manuscript, especially to which studies reflect to. The main idea of the Communication must be highlighted and mentioned.
Response: The main idea has been highlighted in the new Abstract on Page 1.
The main idea of this paper is to reveal the activity dip caused by crystal surface roughness when temperature varies. The vibration and frequency-temperature characteristics of quartz crystal plate with rough surfaces are analyzed in this paper. The resonant frequencies, frequency-temperature curves and vibration modes of the quartz crystal plates are obtained by free vibration analysis with the partial differential equation (PDE) module of COMSOL Multiphysics software. It is shown that the crystal surface roughness reduces the operating frequency of the resonator and causes mode coupling, which is the primary reason for activity dip in the resonator when subjected to temperature variations. These findings provide insights into the occurrence of activity dip and suggest that adjusting the aspect ratio of the resonator can effectively prevent anomalous frequency shifts. For forced vibration analysis, the admittance response and phase response curve of the quartz crystal plate were calculated through the piezoelectric module. It is shown that surface roughness decreases the admittance and phase values of the resonator. When the surface of the crystal is rough, due to structural asymmetry, the positive and negative charges generated by the piezoelectric effect cannot be completely balanced, resulting in the generation of ripples in the admittance and phase values, which is not conducive to the operation of the resonator in the circuit.
- The reflection fot the cited imtes newelty is not unified in the ‘Introduction’ section. In some cases the Author(s) present some studies, in other results and, respectively, for further some weak or strong parts in the current state of knowledge. Please try to refer to each references with the same wages, critical and pointing out some deficiencies.
Response: Necessary introductions about previous works have been added on Page 2.
- The citation of various scholars is not unified as well. Some are referenced by surnames, some items by number, like ‘in Ref. [4]’, line 35. It is not written in a clear way.
Response: This reference citation has been revised on Page 1.
- Each of the cited paper should be introduced separately, respectively to the ‘Inbtroduction’ section again, e.g., [4-7], line 39, [8-14], line 43 or [15-20], line 49. In the last exemple nowelty and critical review should be placed for each of the researchers separately, indicating their progress in the studies.
Response: Considering that the research content and method of these references are similar, it is inappropriate to list them one by one.
- The conclusions received from the ‘Introduction’ section are not derived from the critical review of the literature but, respectively, by the introducing some already finished studies. Proposal of further studies cannot be justified only by previous results but the lack in the current stage of knowledge. From that matter, the further studies must be fulfilling some current requirements, not only another similar studies.
Response: The Introduction Part has been improved according to the reviewer’s suggestion on Page 2.
- All of the equations placed in the second section, titled ‘Theoretical Analysis of Quartz Crystal Plate’, are not introduced as newly proposed but are not referenced. It is not clear what is novelty and what was already presented. The Author(s) should separate, or even highlight, what they suggest and propose, respectively. From that point and comments no. 4 and 5, the motivation, even exist, is lost.
Response: Equations from previous work have been cited in the revised manuscript, which are labeled in yellow on Pages 3 and 4. Additionally, the novelty and motivation of this article has been demonstrated in the fourth paragraph of Introduction.
- Considering the above (previous) comment, if Author(s) present any modification of the mathematical formulas, it must be indicated. Currently, the reader is lost and not receiving what the Author(s) are trying to convey.
Response: For clearly demonstrate the aim of mathematical formulas, some explanations have been added on the beginning and end of Section 2, such as the revisions on Pages 3, 4 and 5.
- In the third section, considering especially 3.1 and 3.2 subsetions, there is no discussion in fact. Any disadvantages and weak points of the studies presented is not introduced. What about the time of computation? Omly some general discussion problems, like in ‘The positive and negative charges generated by the piezoelectric effect cannot completely cancel out, leading to ripples in the admittance and phase curves at the corresponding positions in Figures 6a and 6b, respectively.’, lines 231-233, are placed. Please try to emphasize the advantages and disadvantages of the received results in a more clear way.
Response: Actually, the activity dip caused by surface roughness is revealed with the aid of PDE module in Section 3.1, with the reason interpreted. For quantitatively validating the results obtained and qualitatively verifying the mechanism revealed, the forced vibration is conducted in Section 3.2 by using another module of COMSOL Multiphysics software. For clearly understanding, a short interpretation has been added on Page 7.
- The ‘Conclusion’ section is not acceptable that, respectively, should be divided into separated and numbered gaps. The main (final) proposal must be highlighted and separated from thoe more in detail. Currently, this section is compose by various information not completely reflecting to the presention in the body part of the manuscript.
Response: The conclusions section has been improved according to the reviewer’s suggestion (Page 11).
- The full DOI links should be added to make the reader fast possibility to find the primary sources of the previous results.
Response: The full DOI links for the references have been added as required (Pages 11, 12 and 13).
Reviewer 3 Report
This manuscript investigates the effect of surface roughness on the vibration characteristics of AT cut quartz resonators. The authors demonstrated the effect of surface roughness on the temperature stability and mode coupling by simulation method. The simulation method may be interesting for related researchers. As a communication paper, it may be acceptable by giving more discussions on the following issues.
1) Why do the authors select current Hm values? How does the absolute Hm value affect the vibration characteristics when the hq changes?
2) In Fig. 9, the results of (a) and (b) are greatly different dependent on the value of lq/hq. It seems that Fig. 9(b) shows better performance for Hm=0, so what is the authors' suggestion for resonator designers?
Author Response
1) Why do the authors select current Hm values? How does the absolute Hm value affect the vibration characteristics when the hq changes?
Response: Based on the WA model's assumptions [37], the height of the surface asperities follows a Gaussian distribution. The parametric surface geometric features provided by COMSOL Multiphysics software can generate random surface such as Gaussian random distributions. For quantitatively represent the surface roughness, a factor is utilized in COMSOL Multiphysics software to scale the data in the z direction, which corresponds to the height peak Hm. By adjusting Hm, the height peak of the surface roughness can be modified. Therefore, the height peak Hm is used to characterize the overall roughness of the surface. The surface roughness will have a great influence on the resonator as the plate thickness hq decreases, especially when hq is related to the Hm. The resonant frequency is inversely proportional to the plate thickness. When hq decreases, the influence of surface roughness on the resonant frequency will become more and more evident. The explanations have been added on Page 6.
2) In Fig. 9, the results of (a) and (b) are greatly different dependent on the value of lq/hq. It seems that Fig. 9(b) shows better performance for Hm=0, so what is the authors' suggestion for resonator designers?
Response: Figures 9(a) and (b) demonstrate that quartz crystal resonators with different length/thickness ratios are affected differently by surface roughness. The resonator shows better performance for Hm=0. However, the ideal structure of a smooth surface is not realistic. The results obtained from this paper show that even small changes in surface roughness can have a significant impact on the frequency or displacement of the resonator. This effect is particularly evident when mode coupling occurs, which can lead to activity dip under varying temperatures. To avoid frequency shifts, the length/thickness ratio of the resonator can be altered by grinding or the surface roughness can be improved by polishing. The explanations have been added on Page 10.
Reviewer 4 Report
The work was well designed and could be of interest to broader readers working with the quartz crystal resonator. However, some concerns remain to be addressed.
(1) The admittance of the QCR is dependent on the surface area of the quartz crystal plate. Could the surface area be included in the calculation?
(2) The frequency shift in the QCR is known to be affected by the rigidity of the material attached to the plate surface. How could the roughness of the AT-cut quartz crystal plate be tolerated?
(3) Should the data are applied to the Sauerbrey equation, what is the maximum mass still linearly related to the frequency shift?
(4) In the current real experiment, the temperature is usually kept at room temperature. Could we estimate the optimum temperature?
(5) The current QCR uses either a 5 or 10 MHz base frequency. In this work, why was the base frequency in this simulation far from the commonly used in many works?
(6) The data should be compared with available literature.
Author Response
(1) The admittance of the QCR is dependent on the surface area of the quartz crystal plate. Could the surface area be included in the calculation?
Response: Since a two-dimensional cross-sectional model is established in this paper, the admittance calculated in the COMSOL Multiphysics software is the admittance per unit area of the electrode area.
(2) The frequency shift in the QCR is known to be affected by the rigidity of the material attached to the plate surface. How could the roughness of the AT-cut quartz crystal plate be tolerated?
Response: For a macro resonator with the height peak of surface roughness much smaller than the plate thickness, if the resonant frequency shift caused by the surface roughness is acceptable for engineers, the roughness is tolerated. Additionally, if the length/thickness ratio of resonators is far away from the mode-coupling region, and the activity dip can’t be excited by the surface roughness, it can still be tolerated.
(3) Should the data are applied to the Sauerbrey equation, what is the maximum mass still linearly related to the frequency shift?
Response: The aim of this paper is reveal the activity dip caused by the surface roughness and explain the physical mechanism. The reviewer’s suggestion is interesting, and maybe the quantitative investigation of surface roughness on QCM is carried out utilizing the same method in the near future.
(4) In the current real experiment, the temperature is usually kept at room temperature. Could we estimate the optimum temperature?
Response: We can analyze how temperature affects the vibration characteristics of quartz crystal resonators. For instance, the frequency shift and mode coupling properties when the temperature varies. The aim of this paper is to reveal the activity dip caused by surface roughness when the temperature varies from -55°C to 100°C, or provide some temperature regions without activity dips.
(5) The current QCR uses either a 5 or 10 MHz base frequency. In this work, why was the base frequency in this simulation far from the commonly used in many works?
Response: As a case for numerical simulation, a crystal plate with the thickness 1.2mm is adopted. With the miniaturization and high-frequency requirements of quartz crystal sensors, microscopic devices are made and applied. The present method is not limited by working frequency. For other resonators with different frequencies, the present method is still available, as long as the plate thickness is altered, because the resonant frequency is inversely proportional to the plate thickness.
(6) The data should be compared with available literature.
Response: It is difficult to compare the present results with experimental results in previous literatures. Firstly, fabricating a plate with a specific surface roughness is impossible. Additionally, limited by experimental conditions, the temperature is usually kept at room temperature. Despite of those, the numerical results calculated in free vibration analysis have been validated via comparing resonant frequencies and displacement distributions of forced vibration by utilizing different models.
Round 2
Reviewer 1 Report
This concerns that we proposed have been addressed carefully. Now I'm sure that this manuscript can be accepted for publication in Sensors.
Reviewer 2 Report
The paper was improved and can be accepted in the current, revised form.